# Emotional Eating as a Mediator in the Relationship between Dietary Restraint and Body Weight

**DOI:** 10.3390/nu15081983

**Published:** 2023-04-20

**Authors:** Yaqi Wang, Mandy Ho, Pui-Hing Chau, Susan M. Schembre, Daniel Yee Tak Fong

**Affiliations:** 1School of Nursing, The University of Hong Kong, Hong Kong, China; u3008240@connect.hku.hk (Y.W.); phchau@graduate.hku.hk (P.-H.C.); dytfong@hku.hk (D.Y.T.F.); 2Department of Oncology, Georgetown Lombardi Comprehensive Cancer Center, Georgetown University, Washington, DC 20007, USA; ss4731@georgetown.edu

**Keywords:** eating behavior, dietary restraint, emotional eating, external eating, body weight, adults, mediation

## Abstract

This study aimed to assess the relationships between routine and compensatory restraints and body mass index (BMI), as well as to explore the mediating role of emotional and external eating in the relationships between routine and compensatory restraints and BMI. Chinese adults aged ≥18 years with different weight statuses were invited to fill out an online questionnaire. Routine and compensatory restraints and emotional and external eating were assessed using the validated 13-item Chinese version of the Weight-Related Eating Questionnaire. Mediation analyses tested the mediation effects of emotional and external eating on the relationship between routine and compensatory restraints and BMI. In total, 949 participants (26.4% male) responded to the survey (mean age = 33 years, standard deviation (SD) = 14, mean BMI = 22.0 kg/m^2^, SD = 3.8). The mean routine restraint score was higher in the overweight/obese group (mean ± SD = 2.13 ± 0.76, *p* < 0.001) than in the normal weight (2.08 ± 0.89) and underweight (1.72 ± 0.94) groups. However, the normal weight group scored higher in compensatory restraint (2.88 ± 1.03, *p* = 0.021) than the overweight/obese (2.75 ± 0.93) and underweight (2.62 ± 1.04) groups. Routine restraint was related to higher BMI both directly (β = 0.07, *p* = 0.02) and indirectly through emotional eating (β = 0.04, 95% confidence interval (CI) = 0.03, 0.07). Compensatory restraint was only indirectly related to higher BMI through emotional eating (β = 0.04, 95% CI = 0.03, 0.07).

## 1. Introduction

Obesity is a major public health concern worldwide. According to the World Health Organization (WHO), more than 650 million adults were obese in 2016 [1]. China has the largest number of obese adults, with approximately 46% of adults being overweight or obese [2]. More than one in seven individuals were obese in China in 2018 [3], and the prevalence of overweight and obesity is predicted to increase to 690 million by 2030 [4]. Obesity is linked to reduced life expectancy and quality of life [5,6]. Moreover, it is a major risk factor for non-communicable diseases (NCDs), such as type 2 diabetes mellitus (T2DM) and cardiovascular diseases [1]. It accounted for 17.7% of all-cause deaths and 20.9% of the global burden of NCDs’ disability-adjusted life years (DALYs) in 2017 [7]. Finding effective weight control strategies is a priority in obesity research. A positive energy balance, which occurs when energy intake exceeds energy expenditure, is the main driver of obesity [1,8]. Energy intake is partially regulated by perceived sensations of hunger and fullness [9,10]. However, what and how much we eat is not solely determined by physiological demands. Eating behaviors, including food choices and the amount of food consumed, are also influenced by psychological and social factors [9].

Previous studies [11,12,13] showed that the psychological dimensions of eating behaviors could impact food choices and food consumption, leading to obesity. Emotional eating, external eating and dietary restraint are the three main constructs of psychological eating behaviors. Emotional eating refers to overeating in response to negative emotions such as depression and anxiety [14]. A review of emotional eating showed that a higher level of emotional eating is positively related to weight gain [15]. External eating refers to eating in response to external food-relevant cues, such as sight, smell and taste, rather than internal hunger or satiety state [16]. External eating was found to be significantly associated with higher body mass index (BMI) [17]. In brief, it is known that emotional and external eating are both positively associated with high energy intake and obesity [11,12,17,18].

Dietary restraint refers to the perceived restriction or control of food consumption [19]. Weight loss programs usually require participants to use dietary restraints to restrict their energy intake to promote weight loss. However, the relationship between dietary restraint and body weight has been inconsistent in the current literature. Some cross-sectional studies have reported that dietary restraint is positively related to higher body weight [20,21], and other studies have found that people with higher body weight tend to have lower levels of dietary restraint [18,22,23]. This may be because dietary restraint includes different constructs: routine and compensatory restraints.

Routine restraint refers to the perceived strict and regular restriction of energy intake [24]. Compensatory restraint refers to a more flexible restriction of energy intake, particularly following an episode of overeating [24]. There is mixed evidence on the relationship between routine and compensatory restraints and weight control. A cross-sectional study of multi-ethnic (white, Asian/Asian mixed and native Hawaiian/Pacific Islanders) adults (aged 18–81 years) found that only compensatory restraint was positively associated with weight loss [25]. However, a longitudinal (12-month) study demonstrated that both routine and compensatory restraints were negatively related to weight gain in women enrolled in a weight loss trial [26]. Therefore, it is unknown whether dietary restraint is beneficial for weight loss.

Additionally, a recent study reported that emotional and external eating could mediate baseline dietary restraint and a four-year increase in BMI in female patients with newly diagnosed T2DM [27]. This finding suggests that dietary restraint may be related to long-term weight gain through emotional and external eating. However, no study has tested whether emotional or external eating mediates the relationship between routine and compensatory restraints and BMI.

Clarifying the relationship between dietary restraint and BMI could provide insights into the development of appropriate and more effective weight control strategies to combat obesity epidemics. Therefore, this study aimed to explore the relationship between different constructs of dietary restraint (i.e., routine and compensatory restraints) and BMI among Chinese adults, as well as to assess whether emotional and external eating are the mediators between routine and compensatory restraints and BMI.

## 2. Methods

### 2.1. Study Design

This was a cross-sectional study, and participants were recruited using convenience sampling via social media. Before conducting the study, ethical approval was obtained from the Institutional Review Board of the University of Hong Kong/Hong Kong West Cluster (reference number: UW19-122) and the Hospital Authority in Hong Kong.

### 2.2. Participants

The participants were aged ≥18 years (by self-report) with different weight statuses (underweight, normal weight, overweight and obese) and were able to read Chinese in order to complete an online questionnaire. Individuals were not eligible if they self-reported that they (1) were pregnant or lactating during the investigation; (2) had a history of or a current clinical diagnosis of eating disorder (such as anorexia nervosa); or (3) required dietary restrictions, owing to health conditions, such as diabetes mellitus, hypertension, cancer or food allergies.

### 2.3. Data Collection and Measurements

Data were collected using an online questionnaire via Qualtrics, which included demographic data and self-reported height and weight and the psychological aspects of eating behaviors: routine and compensatory restraints and emotional and external eating, respectively.

Routine and compensatory restraints and emotional and external eating were assessed using the 13-item Chinese version of the Weight-Related Eating Questionnaire (WREQ-C). The validity and reliability of the WREQ-C were examined among Chinese adults [28]. Compared to the other two commonly used tools for assessing the psychological aspects of eating behavior, that is, the Three-Factor Eating Questionnaire (TFEQ) [19] and the Dutch Eating Behavior Questionnaire (DEBQ) [29], the WREQ-C is a shorter scale that assess the two distinct types of dietary restraint: routine and compensatory restraints. The original English version of the 16-item WREQ has been shown to have convergent validity with the TFEQ [26], TFEQ18 [25] and DEBQ [25]. The 13-item WREQ-C consists of three items for routine restraint, three items for compensatory restraint, four items for emotional eating and three items for external eating. Each item was scored between 1–5 (1 = not at all, 2 = sometimes, 3 = half of the time, 4 = most of the time, 5 = always). A higher score indicates a higher level of eating behavior. The final score for each construct was the average of the subscale items. To improve the questionnaire completion rate, responders were reminded to answer skipped or missed questions before proceeding to the next step.

BMI (kg/m^2^) was calculated according to the participants’ self-reported height and weight. Weight status was categorized using the Asian- BMI classification from the WHO [30]. A BMI of <18.5 kg/m^2^ is underweight. Normal weight was defined as a BMI between 18.5 and 22.9 kg/m^2^. A BMI between 23 and 24.9 kg/m^2^ indicates overweight. A BMI of 25 kg/m^2^ or higher was considered obese.

### 2.4. Sample Size Determination

A percentile bootstrap was used in PROCESS macro (version 4.1) to conduct the mediation analysis. According to the guidelines on sample size for mediation analysis [31], a sample size of ≥558 is required to achieve 80% statistical power in a percentile bootstrap when the mediation effect is small.

### 2.5. Statistical Analysis

We used SPSS Statistics for Windows (version 25.0, IBM Corp, Armonk, NY, USA) and PROCESS macro to perform data analysis. Statistical significance was set at a *p* value < 0.05. Descriptive statistics were used to summarize participant characteristics. Continuous and categorical variables are presented as means ± standard deviations (SD) and proportions, respectively. Pearson’s correlation coefficient was used to assess the relationship between routine and compensatory restraints and emotional and external eating with BMI. Independent sample *t*-tests and one-way analysis of variance were conducted to compare the scores of routine and compensatory restraints and emotional and external eating in males and females and individuals of different age groups (18–40 years vs. ≥40 years) and different weight statuses (underweight vs. normal weight vs. overweight/obese). In addition, sensitivity analysis was conducted by excluding older adults (aged >60 years) to compare the scores of routine and compensatory restraints and emotional and external eating between young and middle-aged adults (18–39 years vs. 40–60 years). The PROCESS macro of SPSS was developed by Hayes [32] to perform mediation and moderation analyses. Model 4 in the PROCESS was used to test whether emotional and external eating could mediate the relationship between routine restraint (independent variable) and BMI (dependent variable) and the relationship between compensatory restraint (independent variable) and BMI (dependent variable), respectively. Bootstrapping with 5000 samples was used, and sex and age were controlled for in the models. Mediation effects were calculated using the percentile bootstrap, and the indirect effect was considered significant when the 95% bootstrap confidence intervals (95% CI) did not contain zero [32].

## 3. Results

In total, 949 participants (73.6% females) aged 18–72 years (mean age = 33 years, SD = 14) were included. The participants’ characteristics are listed in Table 1. Approximately half (53.5%) of the participants were normal weight, 31.8% were overweight or obese, and 14.6% were underweight. Most participants (88.2%) had tertiary or higher educational attainment (males: 87.3%, females: 88.5%). About half (48.3%) of the participants had full-time jobs. Moreover, 71.6% of the participants had a monthly household income of more than HKD 20,000 (≈USD 2548) (males: 74.9%, females: 70.4%).

The mean routine restraint score was significantly higher in the overweight/obese group (mean ± SD = 2.13 ± 0.76, *p* < 0.001; Table 2) than in the normal weight (2.08 ± 0.89) and underweight (1.72 ± 0.94) groups. However, the normal weight group (2.88 ± 1.03, *p* = 0.021; Table 2) scored higher in compensatory restraint than the overweight/obese (2.75 ± 0.93) and underweight (2.62 ± 1.04) groups. The emotional and external eating scores did not differ between the underweight, normal weight and overweight/obese groups. The scores for routine and compensatory restraints and for emotional and external eating were significantly higher in females than in males (Table 2). There was no significant difference in the routine and compensatory restraint scores between the participants aged below and above 40 years, although young adults (<40 years old) had higher emotional (2.06 ± 0.97, *p* = 0.002) and external eating (2.62 ± 0.86, *p* < 0.001) scores than middle-aged and older adults (1.85 ± 0.83, 2.27 ± 0.76, respectively) (Table 2).

The results of the sensitivity analysis (by excluding participants aged over 60 years) were similar to the main analyses. When excluding older adults (*n* = 32), there was no significant difference between young and middle-aged adults in routine (young adults: 2.05 ± 0.93; middle-aged adults: 2.03 ± 0.67, *p* = 0.724) and compensatory restraints scores (young adults: 2.81 ± 1.00; middle-aged adults: 2.77 ± 1.02, *p* = 0.588), but young adults scored higher in emotional (2.06 ± 0.97, *p* = 0.015) and external eating (2.62 ± 0.86, *p* < 0.001) than middle-aged adults (emotional eating: 1.89 ± 0.93, external eating: 2.30 ± 0.77).

### 3.1. Correlation between Routine and Compensatory Restraints and BMI

Pearson’s correlation analyses showed that routine restraint (*r* = 0.090, *p* = 0.005) and emotional eating (*r* = 0.131, *p* < 0.001; Table 3) were positively correlated with BMI in the total sample. However, there was no significant correlation between compensatory restraint (*r* = −0.100, *p* = 0.763) or external eating (*r* = 0.049, *p* = 0.133) and BMI. Emotional eating was positively correlated with routine restraint (*r* = 0.251, *p* < 0.001), compensatory restraint (*r* = 0.239, *p* < 0.001) and external eating (*r* = 0.497, *p* < 0.001). Similarly, external eating was positively correlated with routine (*r* = 0.151, *p* = 0.001) and compensatory (*r* = 0.190, *p* < 0.001) restraints. In addition, routine restraint was positively correlated with compensatory restraint (*r* = 0.506, *p* < 0.001).

Gender-specific correlations between psychological eating behavior and BMI were observed. In males, routine and compensatory restraints and emotional and external eating were positively correlated with BMI (*r* = 0.186, *p* = 0.003; *r* = 0.136, *p* = 0.031; *r* = 0.235, *p* < 0.001; and *r* = 0.198, *p* = 0.002, respectively). However, in females, only routine restraint and emotional eating were positively correlated with BMI (*r* = 0.100, *p* = 0.009 and *r* = 0.160, *p* < 0.001, respectively). Compensatory restraint (*r* = −0.001, *p* = 0.968) and external eating (*r* = 0.030, *p* = 0.423) were not correlated with BMI in females. In addition, a positive correlation between routine restraint and external eating was only observed in males (*r* = 0.177, *p* = 0.005).

### 3.2. Mediation Analysis between Routine Restraint and BMI

Figure 1 shows the mediation model of routine restraint, emotional and external eating and self-reported BMI with adjustment for age and sex. The overall mediation model explained 19% (R^2^ = 0.19) of the variance in self-reported BMI (F = 75.66, *p* < 0.001). Routine restraint was related to higher levels of emotional eating (β1 = 0.24, *p* < 0.001). In this routine restraint mediation model, the total (β = 0.12, *p* < 0.001), direct (β = 0.07, *p* = 0.02) and indirect effects of emotional eating (β = 0.04, 95% CI = 0.03, 0.07) were all statistically significant. This means that emotional eating was a mediator in the relationship between routine restraint and self-reported BMI. In other words, routine restraint was directly associated with higher self-reported BMI and indirectly associated with higher self-reported BMI through emotional eating behaviors.

As for external eating, although routine restraint is related to higher levels of external eating (β1 = 0.09, *p* < 0.001), external eating is not related to self-reported BMI (β1 = 0.05, *p* < 0.001). In the routine restraint mediation model, the indirect effect of external eating was not significant (β = 0.0050, 95% CI = −0.0016, 0.01). This means that external eating was not a mediator in the positive relationship between routine restraint and self-reported BMI.

### 3.3. Mediation Analysis between Compensatory Restraint and BMI

The mediation model of compensatory restraint, emotional and external eating and self-reported BMI with adjustment for age and sex explained 18% (R^2^ = 0.18) of the variance in self-reported BMI (F = 69.50, *p* ≤ 0.001; Figure 2). Similar to routine restraint, compensatory restraint is positively related to emotional eating (β1 = 0.22, *p* < 0.001). However, in this compensatory restraint mediation model, the total (β = 0.04, *p* = 0.18) and direct (β = −0.01, *p* = 0.64) effects were not significant, but the indirect effect of emotional eating was significant (β = 0.04, 95% CI = 0.03, 0.07). This means that emotional eating is a mediator in the relationship between compensatory restraint and self-reported BMI. Compensatory restraint was not directly related to self-reported BMI but was indirectly correlated with higher self-reported BMI through emotional eating behaviors.

Similar to the mediation model of routine restraint, compensatory restraint is related to higher levels of external eating (β1 = 0.17, *p* < 0.001), and the indirect effect of external eating was not significant (β = 0.01, 95% CI = −0.0029, 0.02). This means that external eating is not a mediator between compensatory restraint and self-reported BMI.

## 4. Discussion

To our knowledge, this is the first study to investigate the relationships and underlying pathways between routine and compensatory restraints and body weight (based on BMI) and to examine whether emotional and external eating mediate the associations between routine and compensatory restraints and self-reported BMI among Chinese adults. We found that individuals with overweight/obesity reported a higher level of routine restraint compared to the normal weight and underweight groups, whereas the self-reported levels of compensatory restraint were higher in the normal weight group than in the underweight and overweight/obese groups. The results of the mediation analyses showed that routine restraint was related to higher self-reported BMI both directly and indirectly (through emotional eating), whereas compensatory restraint was only indirectly related to higher self-reported BMI through emotional eating.

Dietary restraint includes rigid and flexible control of eating behaviors [33,34]. Rigid control refers to strict dieting with “all or nothing” eating approaches. This is related to high disinhibition and is similar to routine restraint. Conversely, flexible control is a flexible approach to control weight in which “fattening food” is allowed in limited quantities. Flexible control is related to low disinhibition and similar to compensatory restraint [24,33,35]. We found that overweight and obese Chinese adults reported a higher level of routine restraint than the normal weight adults. Routine restraint was associated with higher BMI in both males and females. This finding is consistent with those of previous studies, indicating that rigid control was related to higher BMI or weight gain [24,36,37,38]. This can be explained by the restraint theory, which suggests that restrained eaters might lose control and overeat (i.e., disinhibition), which could lead to higher BMI [39,40,41]. Another possible explanation is that individuals with a higher BMI are more likely to restrict eating to control weight. Some large sample sizes and longitudinal studies have shown that a higher baseline BMI is a predictor of higher levels of dietary restraint [42,43]. The findings of our study, together with those of other studies, confirmed that routine restraint might not be an effective approach to weight management.

Importantly, the results of the mediation analyses provided useful insights into the potential mechanisms of dietary restraint and obesity. We found that emotional eating partially mediated the association between routine restraint and BMI. The model suggests that the higher BMI of routine restraint eaters may be explained by emotional eating. This is consistent with the theory of restrained eating, which explains that restrained eaters would overeat because of negative emotions (similar to emotional eating) [44,45,46,47]. In addition, previous experiments have shown that negative emotions could lead restrained eaters to consume more food than non-restrained eaters [45,48].

However, we did not find a significant association between compensatory restraint and BMI. Similar results have been reported in a diverse population [25]. Compensatory restraint is not related to body weight, possibly because this kind of flexible dietary restraint eating is less strict and might offset the consequences of external eating. Additionally, compensatory restraint eaters are allowed to adjust their energy intake, which could avoid the negative effects of rigid dietary restraint, such as disinhibition and overeating [36,37,40,49].

A novel finding of our study is that the mediation model of compensatory restraint and self-reported BMI suggests that compensatory restraint eaters may have a higher BMI, which could be explained by emotional eating. This could explain the inconsistencies in the current literature regarding the relationship between dietary restraint and body weight [18,20,21,22,23]. A previous study assessed the mediation effects of emotional and external eating on restrained eating and change in BMI and reported that dietary restraint was positively associated with increased BMI after four years through subsequent emotional and external eating in female patients with newly diagnosed T2DM [27]. Our study extended this evidence to the general population and confirmed that both routine and compensatory restraints were associated with higher self-reported BMI through emotional eating. Longitudinal studies are needed to investigate the impact of routine and compensatory restraints on weight changes.

Our study showed that the associations between routine and compensatory restraints and BMI differed. Obesity researchers should use appropriate tools to enable assessment of the different constructs of dietary restraint and investigate the role of routine and compensatory restraints in long-term weight loss and weight maintenance. The positive association between routine restraint and BMI suggests that routine restraint may not be an effective way to control body weight. Weight loss programs could teach participants to adopt a more flexible approach to controlling their dietary intakes. Mediation analyses identified that both routine and compensatory restraints could be positively related to BMI through emotional eating. Therefore, clinicians should assess the levels of dietary restraint and emotional eating in overweight/obese participants and incorporate strategies to address emotional eating in all weight loss programs.

Gender-specific correlations between psychological dimensions of eating behavior and BMI were observed in our study. In males, compensatory restraints and external eating were positively correlated with BMI. In addition, a positive correlation between routine restraint and external eating was observed in males. This implies that gender-specific strategies may be needed to address the impact of psychological eating behavior and weight management.

Our study had several limitations. Owing to the nature of the cross-sectional study design, our results could not provide causal inferences between routine and compensatory restraints and BMI. Further studies are warranted to test the causality of the relationships between the WREQ-C subscales and BMI. Moreover, this study was limited by the use of self-reported body weight and height to calculate BMI and estimate weight status. Using measured health metrics to assess the relationship between dietary restraint and body weight is recommended in future studies. Another limitation is that only Hong Kong Chinese adults were included, and they were recruited through convenience sampling in Hong Kong, which may have reduced the generalizability of the findings to other populations. Furthermore, we did not collect data on the menopausal status of participants, which may affect the levels of dietary restraint [50]. However, the results of the sensitivity analysis by excluding participants aged >60 years was similar to the main analyses. Moreover, the data were collected using an online questionnaire; hence, the findings of our study may not be generalized to individuals with lower digital literacy.

To our knowledge, this is the first study to comprehensively investigate the relationships and underlying pathway between routine and compensatory restraints and body weight among a large sample of Chinese adults with different weight statuses. Our results provide insights for resolving the inconsistencies in the relationship between dietary restraint and body weight and enhancing the effects of weight loss programs.

## 5. Conclusions

Routine and compensatory restraints are related to BMI in different ways. Emotional eating mediated the relationship between routine and compensatory restraints and higher BMI, which may explain the positive association between dietary restraint and weight control. Routine restraint is directly related to higher BMI and indirectly related to BMI through emotional eating. However, compensatory restraint is only indirectly related to a higher BMI through emotional eating. Effective strategies should be identified to reduce the impact of routine restraint and emotional eating behaviors on BMI. Future studies should assess the long-term effects of routine and compensatory restraints and the mediating effects of emotional eating on body weight change and maintenance.

## Figures and Tables

**Figure 1 nutrients-15-01983-f001:**
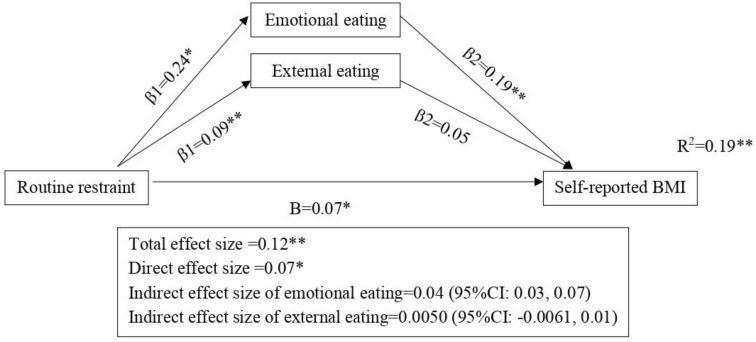
Emotional and external eating as the mediators in the association between routine restraint and self-reported BMI. Standardized beta coefficients (except for R^2^) are shown on the arrows. The models are adjusted for age and gender. * *p* < 0.05 ** *p* < 0.01. Direct and indirect effect size did not add up to total effect size due to rounding.

**Figure 2 nutrients-15-01983-f002:**
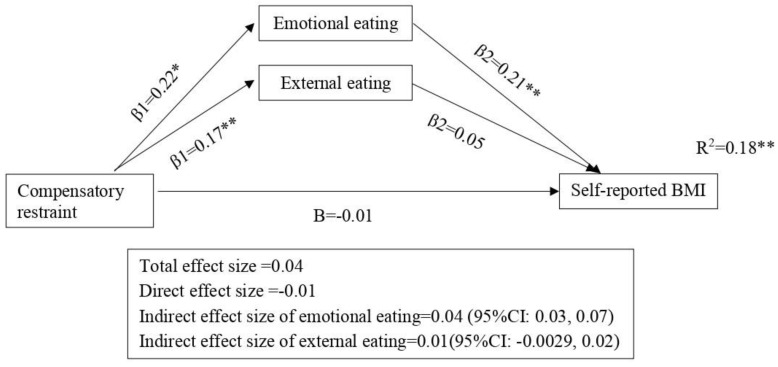
Emotional and external eating as the mediators in the association between compensatory restraint and self-reported BMI. Standardized beta coefficients (except for R^2^) are shown on the arrows. The models are adjusted for age and gender. * *p* < 0.05 ** *p* < 0.01.

**Table 1 nutrients-15-01983-t001:** Characteristics of participants.

	Male (N = 251)	Female (N = 698)	Total (N = 949)
Age (years)	35 ± 15	32 ± 13	33 ± 14
Mean BMI (kg/m^2^)	23.5 ± 3.9	21.4 ± 3.6	22.0 ± 3.8
Weight status ^1^			
Underweight	17 (6.8%)	122 (17.5%) ^4^	139 (14.6%) ^4^
Normal weight	109 (43.4%)	399 (57.2%) ^4^	508 (53.5%) ^4^
Overweight	51 (20.3%)	85 (12.2%) ^4^	136 (14.3%) ^4^
Obese	74 (29.5%)	92 (13.2%) ^4^	166 (17.5%) ^4^
Education level			
Secondary school or below	32 (12.7%)	80 (11.5%)	112 (11.8%)
Tertiary education or above	219 (87.3%)	618 (88.5%)	837 (88.2%)
Occupation			
Full time	144 (57.4%)	314 (45%)	458 (48.3%) ^4^
Part time	7 (2.8%)	27 (3.9%)	34 (3.6%) ^4^
Student/housewives/retired/unemployed	100 (39.8%)	357 (51.1%)	457 (48.2%) ^4^
Monthly Household Income, HKD			
Below HKD 20,000 ^2^	63 (25.1%)	205 (29.6%) ^3^	268 (28.4%)
HKD 20,000 or above	188 (74.9%)	488 (70.4%) ^3^	676 (71.6%)

Values are N (%) for categorical variables and mean ± standard deviation for continuous variables. ^1^ Weight status was categorized using the criteria of the World Health Organization (WHO, 2000): underweight = BMI < 18.5 kg/m^2^; normal weight = BMI 18.5 kg/m^2^ to 22.9 kg/m^2^; overweight = BMI 23 kg/m^2^ to 24.9 kg/m^2^; obese = BMI ≥ 25 kg/m^2^; ^2^ HKD 20,000 ≈ USD 2548 (September 2022); ^3^ 5 missing pieces of data in this category; ^4^ percentages did not add up to 100% because of rounding.

**Table 2 nutrients-15-01983-t002:** The scores of psychological aspects of eating behaviors in individuals with different weight statuses, age and gender.

	Underweight (N = 139)	Normal Weight (N = 508)	Overweight/Obese (N = 302)	*p*-Value	Male (N = 251)	Female (N = 698)	*p*-Value	Young Adults ^1^ (N = 678)	Middle-Aged and Older Adults ^2^ (N = 271)	*p*-Value
Routine restraint	1.72 ± 0.94	2.08 ± 0.89	2.13 ± 0.76	<0.001	1.89 ± 0.80	2.10 ± 0.89	0.001	2.05 ± 0.93	2.03 ± 0.69	0.668
Compensatory restraint	2.62 ± 1.04	2.88 ± 1.03	2.75 ± 0.93	0.021	2.50 ± 0.99	2.91 ± 0.99	<0.001	2.81 ± 1.00	2.78 ± 1.02	0.631
Emotional eating	1.85 ± 0.88	2.00 ± 0.94	2.07 ± 1.03	0.062	1.75 ± 0.88	2.10 ± 0.98	<0.001	2.06 ± 0.97	1.85 ± 0.83	0.002
External eating	2.50 ± 0.85	2.51 ± 0.84	2.55 ± 0.86	0.810	2.38 ± 0.83	2.57 ± 0.85	0.002	2.62 ± 0.86	2.27 ± 0.76	<0.001

^1^ Age: 18 years to 40 years; ^2^ age ≥ 40 years. Values are expressed as mean ± standard deviation. *p* value of one-way ANOVA and independent sample *t*-tests.

**Table 3 nutrients-15-01983-t003:** Pearson’s correlations between psychological eating behaviors and BMI.

	Routine Restraint	Compensatory Restraint	Emotional Eating	External Eating	BMI
Total population					
Routine restraint	1	0.506 **	0.251 **	0.105 **	0.090 **
Compensatory restraint		1	0.239 **	0.190 **	−0.10
Emotional eating			1	0.497 **	0.131 **
External eating				1	0.049
Males					
Routine restraint	1	0.490 **	0.187 **	0.177 **	0.186 **
Compensatory restraint		1	0.171 **	0.359 **	0.136 *
Emotional eating			1	0.534 **	0.235 **
External eating				1	0.198 **
Females					
Routine restraint	1	0.501 **	0.253 **	0.070	0.100 **
Compensatory restraint		1	0.231 **	0.110 **	−0.001
Emotional eating			1	0.476 **	0.160 **
External eating				1	0.030

The results were presented as Pearson’s correlation coefficient. ** *p* < 0.01. * *p* < 0.05.

## Data Availability

The data used for the current study will be available from the corresponding author upon reasonable request.

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
