# Peer review of "Emotional Eating as a Mediator in the Relationship between Dietary Restraint and Body Weight"

_nutrients, 2023, doi:10.3390/nu15081983_

Round 1

Reviewer 1 Report

Congratulations on your manuscript. Some issues rose during reviewing the manuscript.

Rows 76-77 2) had a history of or a current eating disorder diagnosis (such
as anorexia nervosa);

how was is assessed in an online questionnaire? did you just ask the participants? what if they did not know the diagnosis... inclusion criteria should be reformulated.

Graphs are of poor quality, with overlapping lines and need to be redone.

Author Response

Point 1: Rows 76-77 2) had a history of or a current eating disorder diagnosis (such as anorexia nervosa); how was is assessed in an online questionnaire? did you just ask the participants? what if they did not know the diagnosis... inclusion criteria should be reformulated.

Response 1: Thanks for your question. It was assessed by self-report. A screening question was included in the online questionnaire, “have you been medically diagnosed with eating disorder (eg: anorexia, bulimia etc)”. In our study, we excluded the participants if they reported that they have eating disorder. We have revised it to improve clarity. (lines 96-98)

Point 2: Graphs are of poor quality, with overlapping lines and need to be redone.

Response 2: Thanks for your suggestion.  We have reformatted the Figure 1 and 2 to improve the clarity.

Reviewer 2 Report

The presented study assesses the relationships between routine and compensatory restraints and body mass index (BMI), and explores the mediating role of emotional and external eating in the relationships between routine and compensatory restraints and BMI. It is of interest to Nutrition readers and its design and writing make it suitable in its current format, addressing the following issues (minor revision):

It's recommended:

- in 2. Methods (2.2 Participants) anticipate their age range.

- in 2. Methods (2.5 Statistical analysis) explain the sentence: "Model 4 in the PROCESS was used to test whether emotional and external eating could mediate routine restraint (independent variable), compensatory restraint (independent variable), and BMI (dependent variable )." since in the opinion of the reviewer it does not make its meaning clear.

-in 3. Results: grouping people over 40 years of age (middle + older adults) should be justified. Is it necessary? Would the results vary without the numbers corresponding to older people? Do the characteristics of women who have already passed the menopause follow the same patterns?

Author Response

Point 1: Methods (2.2 Participants) anticipate their age range.

Response 1: Thank you for the suggestion. We have revised the statement (lines 94) to improve the clarity.

Point 2: Methods (2.5 Statistical analysis) explain the sentence: "Model 4 in the PROCESS was used to test whether emotional and external eating could mediate routine restraint (independent variable), compensatory restraint (independent variable), and BMI (dependent variable)." since in the opinion of the reviewer it does not make its meaning clear.

Response 2: Thank you for your comment. We have revised the sentence to improve clarity (lines 147-150).

Point 3.1: in 3. Results: grouping people over 40 years of age (middle + older adults) should be justified. Is it necessary?

Response 3.1: Thank you for the question. We compared the levels of eating behaviors between age groups to see if the eating behaviors were related to age. Since age is a confounding factor in the relationship between dietary restraint (routine and compensatory restraints) and self-report BMI, we controlled age in the mediation analyses.

Point 3.2 Would the results vary without the numbers corresponding to older people?

Response 3.2:  Thank you for the question. We have conducted sensitivity analyses by excluding older adults aged over 60 years (n=32), the conclusions remained the same. There was no significant differences between young and middle-aged adults in routine (young adults: 2.05±0.93; middle-aged adults: 2.03±0.67, p=0.724) and compensatory (young adults: 2.81±1.00; middle-aged adults: 2.77±1.02, p=0.588) restraints but young adults scored higher in emotional (2.06±0.97, p=0.015) and external eating (2.62±0.86, p<0.001) than middle-aged adults (emotional eating: 1.89±0.93, external eating: 2.30±0.77). We have added the sensitivity analysis in our manuscript (lines183-189).

Point 3.3 Do the characteristics of women who have already passed the menopause follow the same patterns?

Response 3.3: Thank you for the question. In our questionnaire, we did not collect data on menopausal status. We have revised the manuscript and acknowledged this as a limitation of our study (line 347-349).